# Extranodal Natural Killer/T-Cell Lymphomas: Current Approaches and Future Directions

**DOI:** 10.3390/jcm11102699

**Published:** 2022-05-10

**Authors:** John C. Reneau, Polina Shindiapina, Zachary Braunstein, Youssef Youssef, Miguel Ruiz, Saira Farid, Walter Hanel, Jonathan E. Brammer

**Affiliations:** 1Division of Hematology, Department of Internal Medicine, James Comprehensive Cancer Center, The Ohio State University, Columbus, OH 43210, USA; john.reneau@osumc.edu (J.C.R.); polina.shindiapina@osumc.edu (P.S.); youssef.youssef@osumc.edu (Y.Y.); miguel.ruiz@osumc.edu (M.R.); saira.farid@osumc.edu (S.F.); walter.hanel@osumc.edu (W.H.); 2Department of Internal Medicine, Wexner Medical Center, The Ohio State University, Columbus, OH 43210, USA; zachary.braunstein@osumc.edu

**Keywords:** Epstein-Barr virus, programmed death receptor, brentuximab, daratumumab, extranodal NK/T-cell

## Abstract

Extranodal natural killer/T(NK/T)-cell lymphoma (ENKTL) is a rare subtype of non-Hodgkin lymphoma that typically presents with an isolated nasal mass, but a sizeable minority present with advanced stage disease and have a significantly poorer prognosis. Those with limited disease are standardly treated with chemotherapy and radiation while those with advanced stage disease are treated with L-asparaginase containing chemotherapy regimens. The addition of modern radiation therapy techniques and the incorporation of L-asparaginase into chemotherapy regimens have significantly improved outcomes in this disease, but relapses and death from relapsed disease remain frequent. Given the high rate of relapse, several novel therapies have been evaluated for the treatment of this disease. In this review, we explore the current standard of care for ENKTL as well as novel therapies that have been evaluated for its treatment and the biologic understanding behind these therapies.

## 1. Introduction

Extranodal NK/T-cell lymphoma is a rare subtype of non-Hodgkin lymphoma strongly associated with Epstein-Barr virus (EBV) [1]. The most common cause of this so-called “lethal midline granuloma”, presents almost exclusively in non-nodal sites, usually as a stage IE/IIE disease in the nose, nasopharynx, oropharynx, and upper aerodigestive tract. Other sites such as the skin and gastrointestinal tract can be involved in more advanced stages [2]. Lymph nodes may have secondary involvement but generally are not the primary site of the disease [3]. Given the usual anatomic sites of involvement, most patients present with nasal obstruction or epistaxis; some present with destructive masses that can involve the nose, sinuses, or palate. Approximately three percent of patients with ENKTL present with hemophagocytic lymphohistiocytosis syndrome characterized by high fevers, cytopenia, abnormal liver function tests, and extremely high ferritin levels due to pathologic immune activation [4].

ENKTL is most prevalent in Asia (particularly Japan, China, and Korea) and in Central and South America where EBV infection in early childhood is common [5,6]. It is significantly rarer in other parts of the world including the United States where differences in incidence across various racial groups is evident [7,8]. The median age at diagnosis is 52 years, with only rare reports of this lymphoma in children [9]. Most cases occur in men with a male to female predominance of 2:1 [9].

Demonstration of EBV positivity in malignant cells is required for the diagnosis [10] and likely plays an important role in the pathogenesis of this lymphoma. The cell of origin for ENKTL is usually a natural killer (NK) cell. However, approximately 10% of the time, the malignant cell appears to arise from T-cells, particularly γδ or cytotoxic T-cells [5,11,12,13,14,15]. Morphologically, the tissues involved by ENKTL have extensive ulceration and a diffuse infiltrate of atypical lymphoid cells of varying size admixed with normal-appearing small lymphocytes [1]. The tumor cells often have cytoplasmatic azurophilic granules. A prominent coagulative necrosis is frequently seen in both involved and adjacent normal tissue. An angio-destructive growth pattern is a typical feature, often associated with blood vessel wall fibrinoid necrosis [16].

The typical immunophenotype of ENKTL is similar to that of an NK cell: CD45(+bright), CD2(+), surface CD3(−), cytoplasmic CD3 epsilon(+), CD56(+bright), CD16(−/+), T-cell receptor (TCR) (−), and cytotoxic granule molecules such as granzyme B, TIA-1, and perforin(+) [5]. CD30 is variably expressed, often in larger malignant cells, which can lead to a misdiagnosis of anaplastic large cell lymphoma [17]. TCR genes are in germ-line configuration in most cases, but a small percent have clonal TCR gene rearrangements indicating a T-cell derivation [14,18]; these cases tend to present more often with nodal involvement [15].

The prognosis for ENKTL is variable but it is especially poor for patients with high-risk disease as defined by the prognostic index for natural killer cell lymphoma (PINK-E) [19]. This prognostic index replaced the former NK international prognostic index (NK-IPI) [20] and prognostic factors include age greater than 60 years, stage III or IV disease, distant lymph-node involvement, non-nasal type disease, and a detectable EBV viral DNA titer. Patients in low-risk (zero or one risk factors), intermediate-risk (two risk factor), and high-risk (three or more risk factors) groups have an estimated 3-year overall survival (OS) of 81% (95% CI 75–87%), 55% (95% CI 44–66%), and 28% (95% CI 18–40%), respectively [19]. Given these data, the development of more effective and tolerable novel therapies for this disease beyond currently used cytotoxic chemotherapies is an urgent need.

## 2. Current Standard Therapy: Cytotoxic Chemotherapy and Radiation

### 2.1. Benefit of L-Asparaginase in ENKTL

There is currently no defined standard of care even for front line therapy in limited or advanced stage ENKTL. However, traditional regimens used in many lymphomas such as CHOP (cyclophosphamide, doxorubicin, vincristine, and prednisone) are inadequate due to high expression of P-glycoprotein in malignant cells resulting in chemotherapy resistance [21]. In fact, the OS for CHOP is only 30–40% [22,23]. L-asparaginase is likely the single most important drug for the treatment of ENKTL, especially in the setting of advanced disease. A meta-analysis evaluating the use of L-asparaginase showed that the objective and complete response rates were both improved with the addition of this drug [24] in regimens such as SMILE (steroid (dexamethasone), methotrexate, ifosfamide, L-asparaginase, and etoposide) at over 50% [25] and GELOX (gemcitabine, L-asparaginase, and oxaliplatin) at over 70% (Table 1) [26]. The widespread use of L-asparaginase-based regimens in combination with cytotoxic chemotherapy agents that are not P-glycoprotein substrates has improved survival, although long-term outcomes remain inadequate [27]. In addition, radiotherapy plays an important role for the treatment of limited stage disease [28]. The current treatment paradigms for ENKTL have recently been thoroughly reviewed [29], and we briefly summarize these current standard treatments (see Table 1).

### 2.2. Limited Stage Disease

Radiotherapy (RT) is the most important treatment modality in the treatment of limited stage ENKTL, and patients with localized ENKTL should not receive chemotherapy without RT. For patients receiving RT alone, or receiving sequential chemoradiotherapy, the recommended dosage is 50–54 Gray (Gy), whereas patients who receive concomitant chemoradiotherapy, the RT dose varies depending on the intensity of the chemotherapeutic regimen. Intensity-modulated radiotherapy (IMRT) and volumetric modulated arc therapy (VMAT) are the current standards of care for patients with ENKTL, though novel approaches such as the use of proton therapy, may prove helpful in future studies [35,36]. For patients unfit for chemotherapy, radiotherapy alone may produce cure rates of 50–75%, particularly when risk stratified based on disease characteristics [37].

The current paradigm for the treatment of limited stage disease, i.e., stage I(E) or contiguous stage II(E) generally confined to the nasal and paranasal cavities, consists of RT, combined with platinum containing chemotherapy concurrently or L-asparaginase containing chemotherapy given sequentially or as a “sandwich” with RT. In addition to the gains made in outcomes with the incorporation of L-asparaginase as above, the use of adequately dosed radiotherapy (>50 Gy) with a large enough clinical target volume improves local disease control as well as survival for patients with limited stage disease [28,38].

Commonly used concurrent chemoradiotherapy (CCRT) regimens include RT-2/3DeVIC (radiotherapy, dexamethasone, etoposide, ifosfamide, and carboplatin) and RT-Cis, VIPD (radiotherapy, cisplatin, etoposide, ifosfamide, cisplatin, and dexamethasone). RT-2/3DeVIC concurrently starts radiotherapy and chemotherapy at the same time [30]. RT-Cis, VIPD begins with radiotherapy in combination with weekly cisplatin followed by non-anthracycline containing chemotherapy. In addition to VIPD, several other chemotherapy regimens can be used as treatment after concurrent RT with cisplatin: VIDL (etoposide, ifosfamide, dexamethasone, L-asparaginase) [39], GDP (gemcitabine, dexamethasone, and cisplatin) [40], and MIDLE (methotrexate, ifosfamide, dexamethasone, L-asparaginase, and etoposide) [41].

Nonconcurrent chemoradiotherapy can be given in one of two ways: (1) sequentially with chemotherapy followed by radiation or (2) “sandwich” with a limited number of chemotherapy cycles prior to and following a course of radiation. Modified SMILE is one of the more commonly used regimens, although its use is generally limited to younger, more fit patients given its toxicity profile [42,43]. Other regimens such as P-GEMOX (pegaspargase, gemcitabine, and oxaliplatin) and LVP (L-asparaginase, vincristine, and prednisone), as outlined in Table 1, are generally better tolerated, but direct comparisons of regimens with regards to efficacy are lacking and various single arm trials have significant differences in included patient populations making cross trial comparisons difficult.

For elderly or frail patients with limited stage disease, radiotherapy alone can be used. Early studies have shown a high rate of systemic relapse [44], however, data from more recent times when modern RT techniques were used have shown more favorable outcomes with a 5-year OS of 69.6% and a PFS of 65.1% [37].

For limited stage ENKTL, our current standard of care is to utilize modified SMILE for two cycles, followed by 50 Gy of IMRT to the site of disease in younger, fit patients. For patients unable to tolerate SMILE therapy, we utilize a ”sandwich chemotherapy” approach, with two cycles of GELOX, followed by 50 Gy IFRT, followed by two additional cycles of GELOX.

### 2.3. Advanced Stage and Relapsed Disease

For those with advanced stage disease (stage II (E) with non-contiguous involvement to stage IV) or relapsed, similar chemotherapy regimens as for limited stage disease are often used in the front-line setting. Many of these have L-asparaginase in the regimen as the prior standard of care regimen. CHOP has shown an OS of only 30–40% [22]. In the initial study, SMILE showed an overall response rate (ORR) of 80% with a CR rate of 40% [25]. AspaMetDex (L-asparaginase, methotrexate, and dexamethasone) has been evaluated in a phase II study with an ORR and CR of 78% and 61%, respectively. Additionally, GELOX (gemcitabine, oxaliplatin, and L-asparaginase) has been utilized with a 2-year OS and PFS over 80% [26]. P-GEMOX has been evaluated in a relapsed/refractory population that showed a CR and ORR of 52% and 81%, respectively, and a 3-year OS of 58% [45]. In contrast to limited stage disease, radiation therapy is not used.

Given the generally poor outcomes in this disease, especially for those who present with advanced stage disease, several novel therapies have been evaluated with the hopes of improving outcomes in this disease (Table 2).

## 3. Monoclonal Antibodies

### 3.1. PD1 Inhibitors

The identification of the immune checkpoints was a revolution in understanding the mechanism of cancer evasion and immune escape phenomena. The programmed cell death protein 1 (PD1) is an immune checkpoint protein which manipulates the host immune response by regulating T-cell function. The binding of PD1 ligand (PDL1), present on cancer cells or other non-malignant cells within the immune system, to PD1 on cytotoxic T-cell, leads to T-cell inhibition rendering the cancer cell invisible to the immune system. PDL1 has been found to be overexpressed in EBV + lymphoma (particularly NK lymphoma) as compared with EBV negative lymphomas and is associated with worse prognosis [60,61]. This expression profile has prompted multiple clinical trials to target the PD1-PDL1 axis in ENKTL with the checkpoint inhibitors pembrolizumab [46,61,62] and nivolumab [47].

In a small trial of seven patients with ENKTL with relapsed disease after failing multiple regimens, including L-asparaginase and HSCT, single agent pembrolizumab was given with doses ranging from 100 to 200 mg for a median of 7 total cycles (range 2–13) [62]. Impressive responses were achieved with an ORR of 100%, with CRs and PRs seen in five and two patients, respectively. Grade II skin GVHD occurred in one of the patients who had prior allogeneic HSCT. After a median follow-up of 6 months, the five patients who attained CR remained in CR. PDL1 expression correlated with remission, with strong PDL1 staining in three out five patients with CR and weak staining in one of the two patients with PR. In another trial of ENKTL patients (*n* = 7) with relapsed/refractory disease after failing multiple regimens, including pegaspargase, pembrolizumab was given at 100 mg every 3 weeks [46]. The median number of cycles given in this trial was four. Four patients responded (57% ORR), with two patients attaining a CR. PDL1 was expressed in five out of seven patients (50, 20, 30, 70, and 30%). In a larger study of 14 patients with ENKTL [61], pembrolizumab (100 mg) was given every 3 weeks. An ORR of 44% was seen, with CRs and a PR achieved in five and one patients, respectively. The ORR was higher in those with high PDL1 expression, (67%) as compared with its counterpart with low PDL1 expression (20%). Finally, in a small study with of three patients with ENKTL who failed the SIMPLE regimen of low dose nivolumab [47], a CR was achieved in one patient, while the remaining three patients passed away from infections after showing an initial response. Taken together, these studies show promising activity of PD1-PDL1 axis inhibition in ENKTL, although larger patient numbers with longer follow-up time are needed to assess the durability of responses with single agent PD1 blockade. Combination strategies of other biologic and immune-based therapies with PD1 blockade, including brentuximab and EBV-based therapies, are currently underway and are discussed further in later sections.

### 3.2. Daratumumab

CD38, otherwise known as cyclic ADP ribose hydrolase, is a surface glycoprotein found on the surface of T-cell, B-cells, and natural killer cells. CD38 has a wide range of functions, including modulating cell differentiation, regulating cell recruitment, cytokine release, and regulation of NAD [63]. Over 50% of NK/T-cell lymphomas are CD38+ and CD38 correlates with aggressive behavior and a worse prognosis [64,65]. Daratumumab (Dara) is a monoclonal antibody directed against CD38 which mediates antibody-dependent cell-mediated cytotoxicity and antibody-dependent cell-mediated phagocytosis [66] Emerging evidence is showing promising activity of Dara in ENKTL due to its CD38 positivity [48,67]. A case report of a patient with relapsed/refractory (R/R) ENKTL treated with Dara demonstrated an initial increase in their EBV titers after 6 weeks of treatment. However, EBV titers eventually became undetectable and the patient was able to achieve a remission and remained in remission at 21-weeks of follow-up [48]. For patients with CNS disease, Dara has also been shown to be able to cross the blood-brain barrier to provide a CNS response [49]. Based on these case reports, a multi-center phase 2 study for relapsed/refractory ENKTL with single agent Dara was completed. This showed an overall response rate (ORR) of 25% based on Lugano criteria [50] with all responding patients achieving a partial response (PR) and no patients receiving a complete response (CR). The clinical benefit rate (CR + PR + stable disease) was 44% in this study. The median duration of response was also short at just 55 days. All patients on this study did have at least one adverse event (AE) with the most common being pyrexia (66%), anemia, thrombocytopenia, and transaminitis (28% each), and neutropenia, chills, and headache (25% each), all of which are consistent with previous Dara trials [50]. No studies as of yet have been conducted with Dara in combination with other agents for ENKTL.

### 3.3. Brentuximab

Given the high expression rate in mature T-cell lymphomas and Hodgkin lymphoma [17,68], CD30 expression has also been evaluated in ENKTL and was found to be around 50–70% [17,65,69,70]. A recent large single center study in China showed CD30 expression in a high percentage of ENKTL patients at 47% [69]. The prognostic value of CD30 positivity is not well known, as a study composed of 36 patients showed a favorable outcome with CD30 positivity [71], while another study with 72 patients showed a poor outcome with CD30 positivity [70], and a third study with 22 patients showed no change in OS with CD30 positivity [72]. Brentuximab vedotin (BV) is an antibody targeting CD30 conjugated to the cytotoxic agent monomethyl auristatin E. The phase III ECHELON-2 trial showed that patients treated with BV, cyclophosphamide, doxorubicin, and prednisone (BV-CHP) had a superior outcome as compared with patients treated with CHOP for peripheral T-cell lymphomas, particularly anaplastic large cell lymphomas [73]. A case report from 2015 described a patient with refractory CD30+ ENKTL that achieved a CR with single agent BV [52]. The patient received four monthly cycles of BV with CR, but treatment had to be stopped due to grade 2 dyspnea, and relapse occurred three months later. A second case report described a patient with relapsed ENKTL that achieved CR with the combination of BV and bendamustine [53]. The patient remained in CR at 5-months follow-up. An open-label Phase II trial evaluated the use of BV for R/R high-CD30 expressing non-Hodgkin lymphomas. The subset of patients with ENKTL had an ORR of 29% (2/7, 1 CR, and 1 PR) [51]. Currently, no large-scale studies have been completed evaluating the use of BV for ENKTL specifically. There is a current phase II clinical trial evaluating single agent BV in R/R CD30 low (<10%) mature T-cell lymphomas open to patients with ENKTL (NCT02588651). An additional Phase II pilot trial is currently recruiting patients with relapsed and refractory CD30+ lymphoma to receive BV with or without nivolumab (NCT01703949). Furthermore, a Phase II trial looking at BV and pembrolizumab for patients with recurrent peripheral T-cell lymphoma, including ENKTL, is planned (NCT04795869). Lastly, a phase I/II trial studying BV with methotrexate, L-asparaginase, and dexamethasone for ENKTL is currently recruiting patients (NCT03246750)

## 4. Signaling Pathway Inhibitors

### 4.1. PI3K Pathway Inhibitors

PI3Ks are a family of intracellular signal transduction kinases that phosphorylate the 3′ position of the inositol ring of phosphatidylinositol present in the lipid membrane [74]. This gives rise to PIP3 which binds to the pleckstrin homology domain located in a variety of kinases including Akt, PDK1, and Btk1, leading to their activation which leads to increases in cell metabolism, growth, and cell division. The PI3K are divided into three classes (class I, II, and III) by their structure, regulation, and lipid substrates [75]. The class I kinase isoforms PI3K-δ and PI3K-γ are vital for T-cell functioning [76,77,78]. Duvelisib is an oral dual inhibitor of PI3K-δ and PI3K-γ. In a phase I basket trial to evaluate duvelisib in hematologic malignancies, 15 patients with PTCL were able to be evaluated on treatment. These patients had an ORR of 47% and a median overall survival of 36.4 week, suggesting that duvelisib could potentially be considered for treatment of ENTKL as in PTCL [79]. A further phase I clinical trial looked at 16 patients with R/R PTCL treated with duvelisib and showed an ORR of 50% (3 CR, 5 PR) [80].

A Phase I study is currently recruiting patients to evaluate CC-486 (oral azacitidine) with duvelisib for patients with lymphoid malignancies (NCT05065866). Another study is currently recruiting patients to evaluate ruxolitinib and duvelisib in patients with lymphoma (NCT05010005). Finally, a future trial is planned to evaluate doxorubicin, CC-486, romidepsin, and duvelisib in patients with T-cell lymphoma, including ENKTL. (NCT04639843). While results in ENKTL specifically are limited, it will be important to follow up on these studies to determine if there is a role for PI3K inhibition in ENKTL.

### 4.2. Jak/Stat Pathway Inhibitors

The Jak/Stat pathway is a central signaling pathway within normal T and NK biology, responsible for growth and proliferation in response to cytokine signaling. Not surprisingly, the Jak/Stat pathway has been found to be dysregulated across multiple lymphoma subtypes, including ENKTL. In a series of 84 ENKTL lymphomas sequenced for JAK3 and STAT3, JAK3 mutations were discovered in 7% of cases, while STAT3 mutations were less common, with only one case found to be mutated [81]. However, STAT3 was found to be overexpressed in 51.4% of cases in this same series, suggesting upregulation of this pathway may be a more common method of dysregulation in ENKTL. In a study screening 306 compounds using STAT3 mutant NK lymphoma cells, inhibitors targeting JAK, mTOR, Hsp90, and CDK1 were found to have activity [56]. In a recent biomarker driven study of the Jak1/2 inhibitor ruxolitinib in patients with PTCL (*n* = 45) and MF (*n* = 7) [82], patients were stratified into three different groups based on their dysregulation of the Jak/Stat pathway: (1) activating Jak/Stat mutations, (2) pStat3 overexpression, or (3) no mutations or overexpression. Patients received ruxolitinib at 20 mg twice daily until progression with the primary endpoint of clinical benefit rate (CBR) defined as a CR, PR, or stable disease. The CBR amounts for PTCL patients were 53%, 45%, and 13% in Groups 1, 2, and 3, respectively, demonstrating the importance of Jak/Stat dysregulation in determining response to ruxolitinib inhibition. Although no patients with ENKTL were not enrolled in this study, this does provide further support for investigating Jak/Stat pathway inhibition in ENKTL patients given the relatively high frequency of STAT3 overexpression, as discussed earlier.

## 5. Targeting EBV in the Treatment of ENKTL

EBV is a ubiquitous herpesvirus that infects >90% of adults [83]. EBV primarily infects B lymphocytes and can cause a self-limiting acute infectious mononucleosis characterized by lytic viral replication, expression of lytic proteins such as BZLF1, and potent T-cell-driven immune responses [84,85,86,87]. A minority of EBV-infected B cells may escape the acute immune response. Subsequently, EBV exists in a latent form within B lymphocytes life long, avoiding complete eradication due to the ability of latent infection to evade immune responses [88]. Life-long EBV may undergo periods of lytic re-activation, contributing to infection of T- and NK cells and may drive lymphoproliferative disease development by contributing to malignant transformation of infected cells [89,90]. Latent EBV directly contributes to immune evasion by limiting the expression of immunogenic viral proteins, inducing T-cell senescence, Th1 cell differentiation blockade, and mobilization of regulatory T-cells [91,92,93].

Extranodal NK/T-cell lymphomas arise in immunocompetent individuals and are strongly associated with EBV infection [94]. Expression of EBV-encoded latent membrane proteins 1 and 2 (LMP-1 and LMP-2), EBV nuclear antigen 1 EBNA1, and Epstein-Barr early region (EBER) RNA has been observed in tumor cells [95,96,97,98,99]. Quantification of circulating EBV DNA is used as a surrogate marker of lymphoma burden, with EBV DNA observed at diagnosis, becoming undetectable at remission and remaining elevated in refractory disease [100]. The level of circulating EBV DNA correlates with patient prognosis, response to therapy, and outcomes [101,102,103,104]. Latent EBV infection is thought to contribute to the pathogenesis of ENKTL. Silencing EBNA1 expression using RNA interference inhibited the growth of an EBV-positive ENKTL cell line and was associated with increased expression of p27 protein and subsequent cell cycle arrest [105,106]. Overexpression of the cyclin dependent kinase inhibitor p21, which is frequently observed in ENKTL, may be linked to EBV infection [107]. Furthermore, latent EBV proteins EBNA1, LMP-1, LMP-2 have been implicated in multiple intracellular signaling pathways leading to immortalization and transformation of host lymphocytes (reviewed in [108]).

EBV proteins expressed by ENKTL are relatively weak immunogenic targets, and they present a potential for cytotoxic EBV-specific cytotoxic T lymphocyte (CTL)based therapy for ENTKL treatment [109]. Six ENKTL patients with active disease were treated with LMP-specific CTLs, among them, four patients received T-cells specific to both LMP1 and LMP2, and two patients received LMP2 specific T-cells [57]. Complete responses were observed in four patients and persisted for a median of follow-up period of 3.1 years after T-cell administration [57]. Furthermore, five ENKTL patients that were in remission but considered to be at a high risk of relapse also received EBV-specific T-cells (one subject received LMP2-specific T-cells and five subjects received LMP1- and LMP2-specific T-cells). These subjects all remained in sustained remission after the infusion [57]. No off-target adverse effects were observed. In a subsequent study, five patients with EBER+ ENKTL in remission that had undergone allogeneic stem cell transplant receive donor-derived EBV-specific T-cells targeting LMP1 and LMP2 [110]. Two patients relapsed before 8 weeks after infusion and died of their disease less than 1-year post infusion [110]. One patient relapsed 16 months after infusion. Two patients remained in sustained remission 13 months and 3 years post infusion [110]. Additionally, 1 patient with active disease received LMP1- and LMP2-targeting T-cells and died of disease in 3 months [110]. No toxicities were observed in the ENKTL patients, although one patient with chronic active EBV/hemophagocytic lymphohistiocytosis developed grade 4 hepatic necrosis and one patient developed graft-versus-host disease on this study. Interestingly, T-cell products appeared heterogeneous, and responders generally received higher numbers of LMP2-reactive T-cells than non-responders, suggesting that outcomes may be improved further by enriching the T-cell product for LMP2-specific T-cells [110]. These results were extended further in a phase 2 clinical trial to develop autologous EBV-specific T-cells (baltaleucel T) for treatment and prevention of relapse in advanced, relapsed ENKTL [58]. Autologous polyspecific EBV-targeting T-cells were created by simulation of the patients’ T-cells derived from the peripheral blood with antigen-presenting cells pulsed with Pepmixes derived from EBV targets LMP1, LMP2, BamH1 rightward-open reading frame 1 (BARF1), and EBNA1 [58]. Among 47 patients that enrolled, 15 patients received a median of four EBV-specific T-cell doses, while the rest did not receive the T-cell product due to manufacturing failure, rapid disease progression before the cells could be given, or death [58]. Among the T-cell recipients, 10 patients had active ENKTL at the time of product infusion and demonstrated a 30% complete response (CR) and a 50% overall response rate (ORR) [58]. Five patients had no measurable disease at the time of cell infusion and showed disease progression in three patients and sustained remission in two patients [58]. Taken together, these results suggest that T-cells targeting EBV antigens are well tolerated and capable of inducing remission and preventing relapse in relapsed/refractory ENKTL. Further exploration and optimization of T-cell product manufacture and reactivity against particular EBV epitopes remains a challenge and may contribute to further improvement of outcomes.

Lytic cycle induction using the histone deacetylase inhibitor nanatinostat in combination with the antiviral agent valganciclovir has also been attempted at targeting relapsed/refractory EBV-positive ENKTL [59]. Antiviral agents such as ganciclovir require the presence of functional viral thymidine kinase, BGLF4, a lytic phase protein, for activation [111]. Nanatinostat induces the expression of BGLF4 EBV protein kinase in EBV-positive tumor cells, activating ganciclovir (GCV) via phosphorylation [59]. The Phase 1b/2 VT3996-201 study combined nanatinostat with vanganciclovir for treatment of histologically confirmed EBV-positive lymphomas relapsed/refractory after at least one prior systemic therapy [99]. Among the 55 patients enrolled, nine patients had ENKTL; five patients had PTCL, not otherwise specified; six patients had angioimmunoblastic T-cell lymphoma; and one patient had cutaneous T-cell lymphoma. In 15 patients with T/NK non-Hodgkin’s lymphoma with a median follow-up period of 5.7 months (range 1.9–34.1), an ORR of 60% and CR of 27% were achieved, with a median duration of response of 10.4 months [59]. It is important to note that two patients (ENKTL and PTCL-NOS) in PR and CR, respectively, were withdrawn at 6.7 and 6.6 months, respectively, for consolidation with autologous stem cell transplantation. These results demonstrated promising activity of HDAC inhibitor combined with antiviral agent, especially in T/NK EBV-associated malignancies.

Targeting the effects of EBV on the immune system has also shown potential. EBV is known to induce immune response evasion and was associated with high PD-L1 expression in 15 of 28 tested ENKTL samples [61]. Patients with relapsed/refractory NK/T-cell lymphoma with high PDL1 expression showed a higher response to pembrolizumab (4/6, 67%) as compared with those with low PDL1 expression (1/5, 20%) [61]. This approach warrants further study, perhaps in combination with avenues directly targeting EBV.

## 6. Stem Cell Transplantation in ENKTL

Stem cell transplant (SCT) has been evaluated for the treatment of patients with ENKTL. However, due to the rarity of the disease, and given most patients have limited stage disease (>60%), no prospective, randomized clinical trials exist, so evidence is based primarily from small trials and retrospective studies, mostly in Asian populations. For patients with early stage (IE-IIE) disease, treatments with standard chemoradiotherapy approaches provide cures in most patients, and therefore, SCT is only considered for patients with advanced stage (III-IV) or relapsed/refractory disease.

### 6.1. Role of Consolidative Auto-SCT

There are limited data on the role of consolidative auto-SCT in ENKTL. Consolidative auto-SCT is used routinely in many centers for patients with peripheral T-cell lymphomas (PTCLs) with chemo-sensitive disease, and therefore, this approach has been utilized in patients with ENKTL. In a retrospective study in Asian populations, patients treated without auto-SCT were compared to results from three published series of patients treated with auto-SCT. All patients were matched for NK-IPI between the two groups. In this study, patients in CR at the time of auto-SCT had improved disease-free survival at 5 years (87.3% vs. 67.8%, *p* = 0.027), though this was not observed for those who were not in CR [112]. A key limitation of this study is that these patients were treated in the era prior to routine use of asparaginase with CHOP-based regimens that are now known to be inferior. In a small series of primarily U.S. patients, consolidative auto-SCT improved PFS at 1 (83% vs. 57%) and 3 years (83% vs. 14%), *p* = 0.013 on univariate analysis. This benefit was not confirmed on multivariate analysis, though this study is limited by small numbers [113]. The EBMT reported on results of auto-SCT patients with ENKTL, with 2-year PFS and OS of 33, and 40%, though 57% of patients were not in CR at the time of auto-SCT [114]. Given the limited data available, it is difficult to definitively recommend consolidative auto-SCT for patients with advanced stage ENKTL. However, the available data suggest there may be some benefit for patients in CR after initial treatment who proceed with consolidative auto-SCT.

### 6.2. Role of Allo-SCT

Allo-SCT has been utilized in both the consolidative and salvage setting in ENKTL. Evidence of a graft-versus-lymphoma (GVL) effect in ENKTL comes from numerous studies in which patients attained long-term remissions even with refractory ENKTL [113,115]. In one series of allo-SCT recipients for ENKTL, 1- and 3-year OS was 54% and 39%, respectively, while PFS was 31% at 1 and 3 years. In this cohort, the cumulative incidence of non-relapse mortality (NRM) was 39% at 1 year. In that study, patients receiving any transplant (auto-SCT or allo-SCT) had improved outcomes when transplanted in CR (67% vs. 13% PFS, *p* = 0.002). Interestingly, this series included four patients who received a prior auto-SCT. In a large series from the Center for International Blood and Marrow Transplant Research (CIBMTR), of mixed racial backgrounds (66% Caucasian, 19% Asian ethnicity), 82 patients received allo-SCT of which 30% received up-front allo-SCT, and 60% received salvage therapy after relapse; 45% of the patients were in CR, 30% in PR, and 12% unknown or refractory disease. In this series, 3-year OS and PFS were 34% and 28%, respectively, and the 3-year cumulative incidence of NRM was 30% and relapse mortality was 42%. No relapses were observed beyond 10 months. There was no difference in outcomes in those who received up-front versus salvage allo-SCT, though the favorable outcomes with salvage allo-SCT should be interpreted with caution as many patients will not reach allo-SCT due to refractory disease or rapid progression. These results suggest that allo-SCT is an effective salvage therapy for patients with relapsed/refractory ENKTL, even with just a PR, and can be considered to be a consolidative therapy up-front for highly selected fit patients.

The current data available provides evidence that consolidative auto-SCT may be considered for patients with advanced stage III/IV ENKTL at diagnosis, and a CR after chemotherapy, particularly if they are unfit for an allo-SCT. Allo-SCT should be considered for extremely fit patients in CR1, or those patients who do not attain a CR with initial therapy or with relapsed/refractory disease, but its efficacy is limited by non-relapse mortality.

## 7. Recommendations for the Management Advanced and Relapsed/Refractory ENKTL

Our current approach for patients with advanced and relapsed/refractory ENKTL is to start with L-asparaginase containing chemotherapy. For younger, fit patients, from two to four cycles of modified SMILE, followed by auto-SCT for those in CR is utilized. For less fit patients, ffrom four to six cycles of GELOX chemotherapy is utilized, with consolidative auto-SCT utilized for fit, older patients. The timing and best use of novel agents in ENKTL is not known, and most of these agents are unavailable. For patients who received prior SMILE, GELOX is a good option. For patients that are refractory to chemotherapy, we utilize immunotherapies such as the PD-1 inhibitor nivolumab. Novel agents such a Nstat or EBV-targeting T-lymphocytes are currently unavailable outside of a clinical trial, but at our center these are the primary strategies we use to treat these patients. Allo-SCT should be considered for any patients with multiply relapsed ENKTL, particularly if they failed auto-SCT, even if the patient is only in a partial remission.

## 8. Conclusions and Future Perspectives

ENKTL is a rare disease with relatively poor outcomes, especially for patients with advanced stage disease at diagnosis. Outcomes have significantly improved with the addition of modern radiotherapy techniques and the addition of L-asparaginase into chemotherapy regimens. Despite these improvements, relapses remain frequent and novel therapies such as those discussed here are urgently needed. As we continue to grow in our understanding of the pathogenesis of this disease, we anticipate that novel therapies targeting these drivers will steadily improve the quality of life for patients afflicted with ENKTL.

## Figures and Tables

**Table 1 jcm-11-02699-t001:** Current standard of care therapies for ENKTL.

Arbor Stage	Regimen	Cycles	OS/PFS % (Median Time)	Adverse Effects Most Severe/Most Frequent (N)	Patient Considerations	Evidence (N)
**Concurrent Chemoradiation Therapy (CCRT)**						
IE/IIE	**RT-DeVIC**	50 Gy,3 Cy	78/67 (2-year) 70/63 (5-year)	Nasal skin perforation (1)/eye grade 1/2	No Liver cirrhosis, concomitant malignancy, or CNS involvement	RCT, Yamaguchi et al. [30] JCOG0211 (33)
IE/IIE	**RT-Cis, VIPD**	40 Gy–3 Cy, 3 Cy	86/85 (3-year)	Septic Shock (2)/hematologic toxicity	Adequate hematologic and liver function. EBV DNA (≥64 copies/μL) indicated unfavorable prognosis	RCT, Kim, S.J., et al. [31] (30)
**Asparaginase-Based Chemotherapy/Chemoradiotherapy**						
IE/IIE	**GELOX-RT**	2 Cy, 56 Gy, 2–4 Cy	84/73 (2-year)	Leukopenia (9)/thrombocytopenia (20), decreased fibrinogen (17)	Concomitant malignancies and primary sites other than aerodigestive tract were excluded	Prospective, Wang et al. [26] (20–27 pts were not given pegaspargase due to positive skin test.)
IE/IIE	**LVP-RT “sandwich”**	2 Cy, 56 Gy, 1 Cy	88.5/80.6 (2-year)	Grade 3 RT-dermatitis (5)/RT-mucositis (13)	Extranasal involvement excluded	Jiang et al. [32] (26)
IV/rel/ref	**SMILE**	2 Cy	55/53 (1-year)	Neutropenia (35)	Patients with ischemia/arrythmia excluded, autoHSCT seemed to improve OS/PFS but not statistically significant	Yamaguchi et al. [25] (38)
IE/IIE	**P-GEMOX**	2 Cy, RT, 1 Cy	82/87 (2-year)	Mucositis (11)/anemia (32)	NKIPI score of 0–1, Low EBV load and >54 Gy were favorable predictors of OS/PFS	Wei et al. [33] (35)
II-IV	**DDGP**	6 Cy	68.5/61.8	Multiple organ failure/leukopenia	No other chemo prior to regimen	Zhang et al. [34] (25)

Cis = cisplatin; CNS = central nervous system; Cy = cycle(s); DDGP = cisplatin, dexamethasone, gemcitabine, and pegaspargase; DeVIC = dexamethasone, etoposide, ifosfamide and carboplatin; EBV = Epstein-Barr virus; ENKTL = extranodal NK/T cell lymphoma; GELOX = Gemcitabine, L-asparaginas oxaliplatin; HSCT = hematopoietic stem cell transplant; LVP = L-asparaginase, vincristine, and prednisone; Gy = Gray; N = number; NKIPI = extranodal NK/T cell lymphoma international prognostic index; OS = overall survival; PFS = progression free survival; P-GEMOX = Pegaspargase, Gemcitabine, oxaliplatin; RCT = randomized controlled trial; RT = radiotherapy; SMILE = steroid (dexamethasone), methotrexate, ifosfamide, L-asparaginase, and etoposide; VIPD = etoposide, ifosfamide, cisplatin, and dexamethasone.

**Table 2 jcm-11-02699-t002:** Investigational therapies for ENKTL.

Regimen	Cycles	Response (Time)	AE Most Severe/Most Frequent	Population Considerations	Evidence (N)
**Biologics**					
Pembrolizumab	Varies (2–13)	5 CR, 2 PR (6 months)	Rash (1-AlloHSCT associated)	Relapse to extranasal sites	Li et al. [46] (7)
Nivolumab	Varies (2–9)	3 CR	2 died from infectious etiologies with resolution of extranasal sites, which was attributed to poor condition	L-Asparaginase containing regimens that achieved CR prior to relapse	Retrospective, Chan, T.S.Y., et al. [47] (3)
Daratumumab	6, 4/1–14, (median 2)	2 CR (21 weeks, 6 weeks)/ORR 25%, all achieved PR (49 days)	Thrombocytopenia (8)/pyrexia (19)	Case reports both bad CSF involvement, the first had 21-week PFS, the second died 3 weeks after 4th infusion, Phase II trial	Case reports, Hari et al. [48] and Aeppli [49] (2), Phase II Huang, H., et al. [50]
Brentuximab	Varies, (4, 3) One case was combined with bendamustine	2 CR (4 months, 3 months), Phase II, ORR 29%, PR	Neutropenia	CD30 expression (>0.9%) was found in 56% of cases and significantly higher in extra nasal cases, Though it was not a prognostic OS or PFS factor	Phase II, Kim, H.K., et al. [51] Cohort Expression study, Kawamoto et al. [17] (97) Case reports, Kim, H.K., [52], Poon, L.M., [53] Horwitz et al. [54]
**Small Molecule Inhibitors**					
Vorinostat	1 year	Unknown	Constipation, creatinine elevation	Pediatric patient with other lines of therapy	Case study, McEachron, T.A., et al. [55]
Ruxolitinib					Kuusanmaki et al. [56], Phase II RCT (no ENTKL pts recruitied)
Duvelisib					NCT04639843
**EBV-Targeting Therapy**					
LMP-1/2 CTLs	2 infusions	97% in SR as adjuvant treatment (median 3.1 years); 62% ORR in R/R disease (52% CR)	CNS deterioration, SIRS/none		Bollard et al. [57] (4)
Polyspecific EBV CTLs	median 4 (3–6)	50% ORR, PFS 12 months	None/diarrhea, vomiting		Kim, W.S., [58] (10 salvage cohort, 5 adjuvant cohort)
Nanatinostat with valganciclovir		ORR 60%, CR 27% (median response duration 10 months)			Haverkos [59] (15 EBV-assoc lymphomas, 9 ENKTL)

AE = adverse events; AlloHSCT = allogeneic hematopoietic stem cell transplant; CNS = central nervous system; CR = complete response; CTL = cytotoxic T lymphocyte(s); EBV = Epstein-Barr virus; ENKTL = extranodal NK/T cell lymphoma; LMP = latent membrane proteins; N = number; ORR = objective response rate; PFS = progression free survival; PR = partial response; R/R = relapsed/refractory; RCT = radomized controlled trial; SIRS = systemic inflammatory response syndrome.

## Data Availability

Not applicable.

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
