# Peer review of "Extranodal Natural Killer/T-Cell Lymphomas: Current Approaches and Future Directions"

_jcm, 2022, doi:10.3390/jcm11102699_

Round 1

Reviewer 1 Report

In the article “Extranodal Natural Killer/T-Cell Lymphomas: Current Approaches and Future Directions”, Reneau JC et al did an extensive review on the treatment of extranodal Natural Killer/T-Cell Lymphomas, especially focusing on the novel therapies. The article is generally well organized and the bibliography is accurate. However, some changes could be introduced to improve it.

The authors mentioned that the last most important advance in the treatment of ENKTL has been the introduction of L-asparaginase in the regimens to treat this lymphoma. However, little is said regarding this issue. I would recommend to compare the results of treatment between the pre and post L-asparaginase era, highlighting the most important results achieved with the introduction of this drug.

The abstract should be reviewed. The sentence “…while those with advanced stage disease are treated with L-asparaginase chemotherapy alone.” is not correct as L-asparaginase is usually given in combination with other drugs. Moreover, the sentence “in this review, we will present our current biologic understanding of ENKTL…” should be modified as the article is about treatment, but the etiopathogenesis and biological mechanism of the disease are not reviewed, other than those related with the therapies.

The sentence on lines 39-41 is grammatically incorrect “…the United States here differences…”. Please correct line 46 “approximately 10% of the time”.

On line 78 correct EKTL (ENKTL).

The title of the sections “Biologic therapies” and “Small molecules therapies” sound somehow weird. I would recommend to replace them; the former by “immunotherapy” or “monoclonal antibodies” and the latter by “Signaling pathways inhibitors”.

Author Response

In the article “Extranodal Natural Killer/T-Cell Lymphomas: Current Approaches and Future Directions”, Reneau JC et al did an extensive review on the treatment of extranodal Natural Killer/T-Cell Lymphomas, especially focusing on the novel therapies. The article is generally well organized and the bibliography is accurate. However, some changes could be introduced to improve it.

Thank you for your kind comments.

The authors mentioned that the last most important advance in the treatment of ENKTL has been the introduction of L-asparaginase in the regimens to treat this lymphoma. However, little is said regarding this issue. I would recommend to compare the results of treatment between the pre and post L-asparaginase era, highlighting the most important results achieved with the introduction of this drug.

We thank the reviewer for this comment. In studies, CHOP showed an OS of around 30-40% while L-asparaginase containing regimens showed a much higher OS, including >50% for SMILE and >70% for GELOX. The following has been added to page 5, paragraph 1: “In fact, the OS for CHOP is only 30-40%22,23… A meta-analysis evaluating the use of L-asparaginase showed that the objective and complete response rates were both improved with the addition of this drug24in regimens like SMILE (steroid [dexamethasone], methotrexate, ifosfamide, L-asparaginase, and etoposide) at over 50%25 and GELOX (gemcitabine, L-asparaginase, oxaliplatin) at over 70% (Table 1)26

The abstract should be reviewed. The sentence “…while those with advanced stage disease are treated with L-asparaginase chemotherapy alone.” is not correct as L-asparaginase is usually given in combination with other drugs. Moreover, the sentence “in this review, we will present our current biologic understanding of ENKTL…” should be modified as the article is about treatment, but the etiopathogenesis and biological mechanism of the disease are not reviewed, other than those related with the therapies.

Thank you for this comment. We agree with the above comments and have made changes to the abstract. We have changed the advanced stage treatment sentence to discuss patients being treated with L-asparaginase containing regimens, such as SMILE.

On page 2, the abstract has been changed to read “Extranodal NK/T cell lymphoma is a rare subtype of non-Hodgkin lymphoma that typically presents with an isolated nasal mass, but a sizeable minority present with advanced stage disease and have a significantly poorer prognosis. Those with limited disease are standardly treated with chemotherapy and radiation while those with advanced stage disease are treated with L-asparaginase containing chemotherapy regimens, such as SMILE. The addition of modern radiation therapy techniques and the incorporation of L-asparaginase into chemotherapy regimens have significantly improved outcomes in this disease, but relapses and death from relapsed disease remain frequent. Given the high rate of relapse, several novel therapies have been evaluated for the treatment of this disease; in this review, we will explore the current standard of care for ENKTL as well as novel therapies that have been evaluated for its treatment and the biologic understanding behind these therapies.

The sentence on lines 39-41 is grammatically incorrect “…the United States here differences…”. Please correct line 46 “approximately 10% of the time”.

We thank the reviewer for this comment. This has been corrected to “the United States where differences…” on page 3, paragraph 2.

On line 78 correct EKTL (ENKTL).

The title of the sections “Biologic therapies” and “Small molecules therapies” sound somehow weird. I would recommend to replace them; the former by “immunotherapy” or “monoclonal antibodies” and the latter by “Signaling pathways inhibitors”.

We thank the reviewer for this comment. The section titles have been changed to “Monoclonal antibodies” (page 7) and “Signaling Pathway Inhibitors” (page 10).

Reviewer 2 Report

This review is summarizing available data of the rare extranodal T/NK lymphomas. The reviewer collected all available treatment options, and lists them in a table.

The introduction and background section is good.

I think the concept is good, but I would recommend restructuring this paper to be more helpful for the reader. The early stage disease section is detailed, however no author opinion is there regarding what treatment to choose. 

Also radiation therapy should be expanded, especially for the midline granuloma. However data is limited, but this is a very special field where the very focused proton beam therapy can be applied. This should be mentioned here as well.

The advanced disease treatment option section is not acceptable. It should be expanded, referencing the treatments provided in the table. The reader should have a view as what treatment is recommended, for what condition. Also even novel treatments are listed below, the author should comment on the decision as to choose chemo or novel treatment.

The best part is the novel treatments. They have essential role, but given the small patient number, there is no evidence so far as to what option is the better. This section is balanced, detailed.

I have a feeling that the author wanted to deliver this data in the paper. However, to match the title and clinical need, the conventional treatment options should be expanded to this level as well.

I recommend restructuring the paper, expanding the chemo section, especillay the advanced disease section. 

Author Response

This review is summarizing available data of the rare extranodal T/NK lymphomas. The reviewer collected all available treatment options, and lists them in a table.

The introduction and background section is good. Thank you for your comments.

I think the concept is good, but I would recommend restructuring this paper to be more helpful for the reader. The early stage disease section is detailed, however no author opinion is there regarding what treatment to choose.

We thank the reviewer for the comment. We have included a paragraph regarding the current approach at OSU for limited/early staged ENKTL as below:

For limited stage ENKTL, our current standard of care is to utilize modified SMILE for two cycles, followed by 50 Gy of IMRT to the site of disease in younger, fit patients. For patients unable to tolerate SMILE therapy, we utilize a ‘sandwich chemotherapy’ approach, with two cycles of GELOX, followed by 50 Gy IFRT, followed by two additional cycles of GELOX.

Also radiation therapy should be expanded, especially for the midline granuloma. However data is limited, but this is a very special field where the very focused proton beam therapy can be applied. This should be mentioned here as well.

We thank the reviewer for the comment. While XRT is largely out of the scope of this review, and has been reviewed elsewhere, we have added a paragraph on essential principles to the limited stage disease section as below:

“Radiotherapy (RT) is the most important treatment modality in the treatment of limited stage ENKTL, and patients with localized ENKTL should not receive chemotherapy without RT. For patients receiving RT alone, or receiving sequential chemo-radiotherapy, the recommended dosage is 50-54Gy, whereas patients who receive concomitant chemoradiotherapy, the RT dose varies depending on the intensity of the chemotherapeutic regimen. Intensity-modulated radiotherapy (IMRT) and volumetric modulated arc therapy (VMAT) are the current standard of care for patients with ENKTL, though novel approaches such as the use of proton therapy, may prove helpful in future studies.{Liu, 2017 #256}{Wang, 2012 #254} For patients unfit for chemotherapy, radiotherapy alone may produce cure rates of 50-75%, particularly when risk stratified based on disease characteristics.{Yang, 2015 #253}”

The advanced disease treatment option section is not acceptable. It should be expanded, referencing the treatments provided in the table. The reader should have a view as what treatment is recommended, for what condition. Also even novel treatments are listed below, the author should comment on the decision as to choose chemo or novel treatment.

We thank the reviewer for this comment and have updated/expanded this section to include multiple regimens, including SMILE, AspaMetDex, GELOX, and P-GEMOX. We have also included the response to CHOP as a reference. The following paragraph is updated:

“For those with advanced stage disease (stage II(E) with non-contiguous involvement to stage IV) or relapsed, similar chemotherapy regimens as for limited stage disease are often used in the front-line setting. Many of these have L-asparaginase in the regimen as the prior standard of care regimen, CHOP, showed an OS of only 30-40%22. In the initial study, SMILE showed an overall response rate (ORR) of 80% with a CR rate of 40%25. AspaMetDex (L-asparaginase, methotrexate, dexamethasone) has been evaluated in a phase II study with an ORR and CR of 78% and 61% respectively. Additionally, GELOX (gemcitabine, oxaliplatin, and L-asparaginase) has been utilized with a 2-year OS and PFS over 80%26. P-GEMOX has been evaluated in a relapsed/refractory population that showed a CR and ORR of 52% and 81% respectively, and a 3-year OS of 58%40. In contrast to limited stage disease, radiation therapy is not used. “

We have also added a new section at the end on our recommendations for the management of advanced/relapsed/refractory ENKTL:

“Our current approach for patients with advanced and relapsed/refractory ENKTL is to start with L-asparaginase containing chemotherapy. For younger, fit patients, 2-4 cycles of modified SMILE, followed by auto-SCT for those in CR is utilized. For less fit patients, 4-6 cycles of GELOX chemotherapy is utilized, with consolidative auto-SCT utilized for fit, older patients. The timing and best use of novel agents in ENKTL is not known, and most of these agents are unavailable. For patients who received prior SMILE, GELOX is a good option. For patients that are refractory to chemotherapy, we utilize immunotherapies such as the PD-1 inhibitor nivolumab. Novel agents such a Nstat or EBV-targeting T-lymphocytes are currently unavailable outside of a clinical trial, but at our center these are the primary strategies we use to treat these patients. Allo-SCT should be considered for any patients with multiply-relapsed ENKTL, particularly if they failed auto-SCT, even if the patient is only in a partial remission.”

The best part is the novel treatments. They have essential role, but given the small patient number, there is no evidence so far as to what option is the better. This section is balanced, detailed.

Thank you for your comments

I have a feeling that the author wanted to deliver this data in the paper. However, to match the title and clinical need, the conventional treatment options should be expanded to this level as well.

I recommend restructuring the paper, expanding the chemo section, especillay the advanced disease section.

Thank you-this has expanded as above.

Round 2

Reviewer 1 Report

No more comments to add after the second review.

Reviewer 2 Report

The revised manuscript is acceptable.